# Molecular Mechanisms of Na-Cl Cotransporter in Relation to Hypertension in Chronic Kidney Disease

**DOI:** 10.3390/ijms24010286

**Published:** 2022-12-23

**Authors:** Lijuan Liang, Tatsuo Shimosawa

**Affiliations:** Department of Clinical Laboratory, International University of Health and Welfare, Chiba 286-8520, Japan

**Keywords:** salt, WNK, salt-sensitive hypertension, chronic kidney disease

## Abstract

Chronic kidney disease (CKD) is a common clinical disease with an increasing incidence, affecting 10 to 15% of the world’s population. Hypertension is the most common and modifiable risk factor for preventing adverse cardiovascular outcomes in patients with CKD. A survey from developed countries shows that 47% of hypertensive patients over the age of 20 have uncontrolled blood pressure (BP), and the control rate is even lower in developing countries. CKD is both a common cause of uncontrolled hypertension and a risk factor for altered sequelae. In particular, studies have demonstrated that abnormal blood-pressure patterns in CKD patients, such as non-dipping-blood-pressure patterns, are associated with a significantly increased risk of cardiovascular (CV) disease. The distal convoluted tubule (DCT) is a region of the kidney, and although only 5–10% of the sodium (Na^+^) filtered by the glomerulus is reabsorbed by DCT, most studies agree that Na-Cl cotransporter (NCC) in human, rabbit, mouse, and rat kidneys is the most important route of sodium reabsorption across the DCT for maintaining the homeostasis of sodium. The regulation of NCC involves a large and complex network structure, including certain physiological factors, kinases, scaffold proteins, transporter phosphorylation, and other aspects. This regulation network includes various levels. Naturally, cross-talk between the components of this system must occur in order to relay the important signals to the transporter to play its role. Knowledge of the mechanisms regulating NCC activation is critical for understanding and treating hypertension and CKD. Previous studies from our laboratory have investigated the mechanisms through which NCC is activated in several different models. In the following sections, we review the literature on the mechanisms of NCC in relation to hypertension in CKD.

## 1. Introduction

Hypertension and chronic kidney disease (CKD) are two major risk factors for cardiovascular (CV) disease [1]. CKD patients often experience varying degrees of elevated blood pressure; its incidence accounts for about 19.6% to 57.7% of all kidney diseases, and a few patients can develop malignant hypertension [2]. Once hypertension occurs, it can further damage the renal function and encourage blood pressure to continue to rise, forming a vicious circle [3]. A survey from developed countries shows that 47% of hypertensive patients over the age of 20 have uncontrolled blood pressure [3], and the control rate is even lower in developing countries.

Therefore, effective blood-pressure control in hypertensive patients with chronic kidney disease is of great significance to prevent the progression and deterioration of renal disease [4]. The coexistence of hypertension and CKD makes it more difficult to control blood pressure levels. Salt sensitivity is an independent risk factor for cardiovascular disease after hypertension. About half of the hypertensive population and one-quarter of the normotensive population exhibit blood pressure (BP) sensitivity to salt [5]. The pathogenesis of hypertension in CKD is complex, and its occurrence and development are affected by many factors. Sodium-imbalance-increased sympathetic nervous system (SNS) activity and changes in the renin-angiotensin system (RAAS) are some of the key pathogenic mechanisms [6]. An important target of these mechanisms is NCC. NCC activity is highly regulated by a complex signaling network, and these kinases appear to be sensitive to changes in hormonal and physiological environments [7]. They regulate NCC in three main ways, namely, by regulating the level of total protein synthesis and transport and the phosphorylation level of NCC. The change in the total protein level of NCC is a chronic reaction of the body, which can be caused by long-term sodium and chloride imbalances in the body. Transportation and phosphorylation levels are rapid responses, and the two mechanisms are independent of each other. The phosphorylation state of NCC is the main form of its function, and only the phosphorylated NCC expressed on the cell membrane can function [8]. The regulatory mechanisms of NCC phosphorylation are the most frequently studied, such as protein phosphatase-4 to dephosphorylate NCC; serum and glucocorticoid-regulated kinase 1(SGK1), aldosterone can promote NCC phosphorylation (Susa et al., 2012), and so on. It was also reported that NCC phosphorylation is also regulated by NCC ubiquitination, that is, a decrease in NCC ubiquitination leads to an increase in NCC phosphorylation and an increase in NCC phosphorylation downregulates NCC ubiquitination [9]. In this review, the various regulatory mechanisms of NCC in relation to hypertension in CKD are expounded.

## 2. NCC Is Regulated by a Variety of Kinases and Proteins

The total length of the distal convoluted tubule (DCT) of the kidney is less than 0.6 mm, and it can reabsorb 5 to 10% of the glomerular filtrate. The DCT is divided into two parts: the c-initial phase, expressing NCC exclusively in the apical membrane, and the late phase [10], which is the transition part between the DCT1 and the connecting tubule/collecting duct (CNT/CD). The DCT2 apical membrane contains NCC, epithelial sodium channel (ENaC), and renal outer medullary potassium channel (ROMK) [11]. The thiazide-sensitive NCC derived from the bladder of winter flounders is the first electrically neutral co-transporter protein to have been identified at the molecular level [12]. DCT enables different functions of each fragment through specific regulators, such as Ste20-related Proline Alanine-Rich Kinase (SPAK), Cullin 3(CUL3), Kelch-like protein 3 (KLHL3), and neural precursor cell expressed developmentally downregulated gene 4-like (Nedd4-2). The dysfunction of NCC proteins triggers a series of water–sodium imbalances and blood-pressure abnormalities; therefore, NCC’s activity plays an important role in cardiovascular physiology and pathophysiology, and as a microregulator of renal sodium excretion, the relationship between NCC function and the pathological process of hypertension has received continuous attention.

### 2.1. WNK

With-no-lysine kinase (WNK) (rat WNK1) was first cloned by the PCR cloning technique when identifying new members of the mitogen-activated protein kinase kinase (MEK) family [13]. As the main regulator of NCC, its importance is self-evident. WNK kinases play important roles in the control of salt homeostasis and blood pressure. Each of the WNK kinases can interact at the protein level [14]. 

After the identification of WNK1 from rat kidneys, it was soon observed that the WNK kinases consist of WNK1, WNK2, WNK3, and WNK4 [15]. In the kidneys, WNK1 is expressed in two forms: kidney-specific-WNK1 (KS-WNK1) and long (kinase-active) WNK1 (L-WNK1). L-WNK1 whose transcription starts in exon 1, contains the kinase domain and is expressed at low levels in the kidney. The KS-WNK1 is due to its kidney-specific expression starting at the polypeptide sequence encoded by exon 5. KS-WNK contains no kinase domain [16]. L-WNK1 activates epithelial sodium channels and NCC, thus activating sodium reabsorption. KS-WNK1 is thought to inhibit L-WNK1 kinase activity and its effect on NCC through a dominant-negative effect [16].

However, given the recent studies suggesting a role for KS-WNK1 in the response to hypokalemia, KS-WNK1 knockout mice may have a potassium-loss phenotype-induced NCC activation that can be compensated by WNK4 [17]. Furthermore, in X. laevis oocytes, KS-WNK1 expression encourages NCC activation despite the lack of a kinase domain [16]. 

WNK4 appears to be a negative regulator of NCC in some cases. However, sometimes, it is a positive regulator in others [14].

Wild-type WNK4 inhibits NCC activity by interfering with the forward transport of NCC towards the cell membrane; thus, the lysosomal pathway, rather than the clathrin-mediated endocytic pathway, degrades NCC to inhibit the expression of NCC on the cell membrane and reduces the activity of NCC [18].

NCC interacts with the adaptor protein complex 3(AP-3), AP-3 is involved in the transport of lysosomes, and the participation of WNK4 can enhance the interaction between the two, which also indicates that WNK4 can stimulate the interaction between NCC-AP3; therefore, it also supports the transport of NCC to the lysosome [19]. 

As shown by Mu et al. [20], exogenous catecholamine inhibits histone deacetylases containing negative glucocorticoid-responsive elements (nGRE). This stimulates the β2- AR and acetylates the histones. Thus, WNK4/SPAK transcription is reduced and NCC activity is activated, leading to hypertension.

In a study by San-Cristobal et al. [15], the authors used Xenopus laevis oocytes with clones and mutagenesis to show that angiotensin II converts WNK4 from an inhibitor to an activator of NCC. In normal or enlarged intravascular volume, the renin–angiotensin system is suppressed, and WNK4 reduces the amount of NCC in the plasma membrane by inhibiting NCC translocation. In the context of reduced intravascular volume, RAAS is activated and angiotensin II signaling alleviates the inhibition of NCC by WNK4, resulting in increased NCC activity. 

In their study, O’Reilly et al. [21] used male C57BL/6 mice with a specific Na-and-K diet and excess aldosterone via minipump or adrenalectomization to abolish aldosterone production and showed that renal WNK4 expression was upregulated on a high-potassium diet and downregulated on a low-sodium diet, while its mRNA level was not significantly increased with aldosterone treatment, suggesting that WNK4 acts as a regulator to adapt to changes in K^+^ and Na^+^ balance. 

Among other WNK isoforms, WNK3 is also expressed in the kidneys and only in the aldosterone-sensitive distal nephron, whereas WNK2 is not expressed, and the regulation of WNK2-expression levels is currently poorly understood [22]. 

Compared with the inhibitory activity of WNK4, kinase-activated WNK3 is a potent activator of Na-K-Cl cotransporter 2 (NKCC2) and NCC, and kinase-inactivated WNK3 is a potent inhibitor of NKCC2 and NCC activity [23]. 

The evidence for the effect of WNK3 on these transporters and their co-expression in renal epithelial cells suggests that WNK3 is associated with NaCl, water, and blood-pressure stabilization [23]. NCC and NKCC are related kidney-specific transporters that mediate apical NaCl reabsorption in the thick ascending limb and distal convoluted tubule, respectively. WNK3 regulates the activity of these transporters by altering their expression at the plasma membrane [23]. WNK kinases not only directly regulate the ion-transport pathway but also form a signaling complex with each other, and WNK3 can modulate the effects of WNK1 and WNK4 on NCC activity [14]. Increased WNK4 abundance can increase the WNK4/WNK3 molar ratio, thus inhibiting NCC activity both directly (by the effect of WNK4 on NCC) and indirectly (through the effect of WNK4 on WNK3) [24]. 

As a controllable variable resistor amplifier, the WNK kinase complex can adjust the activity of NCC in a cascade according to physiological needs [18].

In their study, Cary R. et al. [25] showed that the WNK bodies that were created by potassium imbalance in DCT were unique KS-WNK1-dependent structures of WNK signaling complexes. They proposed that WNK body formation is associated with the normal physiology of the distal nephron, which is an evolutionarily conserved manifestation of the renal response to potassium stress. Martin N. et al. [26] used genetically engineered mice carrying mutations of WNK-SPAK/OSR1-pathway proteins with varying-K^+^-and-NaCl-content diets to provide evidence that WNK bodies play a key functional role during changes in plasma K^+^ concentration via WNK4-induced SPAK/OSR1 activation.

### 2.2. Ubiquitination

Kinases induce protein phosphorylation at specific threonine, serine, or tyrosine residues. Several membrane proteins, including transporters and channels in the kidneys, are regulated by ubiquitination. The regulation of NCC activity by ubiquitination is an emerging area of interest for researchers, and these ubiquitin (Ub) ligases may have a greater impact on NCC than WNK [8].

For example, Ishizawa et al. [27] demonstrated in low-potassium-fed mice that KLHL3/CUL3-based ubiquitin ligases are involved in low K^+^-mediated NCC activation, a physiological adaptation that reduces distal electrical Na^+^ reloading absorption, preventing the further loss of K^+^ by the kidneys but promoting high blood pressure.

The in vivo and in vitro findings in Penton et al.’s study [28] showed that cyclic adenosine monophosphate (cAMP) increased NCC phosphorylation through the protein kinase A (PKA)-dependent phosphorylation of protein phosphatase 1 inhibitor–1 (I1) and subsequent protein phosphatase 1(PP1) inhibition, and the PKA-mediated phosphorylation of KLHL3 at S433 reduced KLHL3-dependent the ubiquitination and degradation of WNK4, a pathway that may be involved in the physiological regulation of renal sodium processing by cAMP-elevating hormones and may contribute to salt-sensitive hypertension in patients with endocrine dysregulation or sympathetic hyperactivity.

In addition to KLHL3 and CUL3, Nedd4-2 can also stimulate the ubiquitination of NCC. Ronzaud et al. [29] provided evidence that the inactivation of Nedd4L exons 6 to 8 in adult mouse Nedd4L-KO (Nedd4LPax8/LC1) renal tubules do not result in the Liddle-syndrome phenotype associated with elevated ENaC activity; rather, it results in a salt-sensitive Pseudohypoaldosteronism Type II (PHAII)-like syndrome that is characteristic of the upregulation of NCC, increasing blood pressure and hypercalciuria. NEDD4-2 is not important for the regulation of renal ENaC, as low plasma aldosterone leading to the reduced proteolytic cleavage of αENaC may be sufficient to counteract the increased abundance of β- and γENaC in Nedd4LPax8/LC1, and, based on these results, NEDD4-2 appears to target NCC primarily.

### 2.3. Regulation by Several Other Proteins and Kinases

In addition to the regulation of kinases and ubiquitinated proteins, NCC is also regulated by several other proteins and kinases. Ueda K. et al. [30] showed that ENaC and NCC, but not NKCC2, were activated in kidney-specific corticosteroid 11-β-dehydrogenase isozyme 2(Hsd11b2) knockout mice, suggesting that hypokalemia-induced NCC activation augments the renal-ENaC-activation-induced elevation of BP. HSD11B2 converts glucocorticoids to their inactive form and maintains the sensitivity of MR to mineralocorticoids. in the apparent mineralocorticoid excess syndrome Hsd11b2 has a specific variation. In their animal model of salt-dependent hypertension, the antihypertensive effect of potassium supplementation was attributable to urinary sodium excretion [30]. These results strengthen the hypothesis that impaired renal function, rather than vascular dysfunction, contributes to the development of salt-dependent hypertension caused by kidney-specific deletion of the Hsd11b2 gene.

Gholam et al. [31] used Mouse DCT15 cells to confirm that Ca^2+^/calmodulin-dependent protein kinase II (CaMKII) not only indirectly regulates NCC through filamin A but also directly phosphorylates and regulates NCC activity, and the binding between CaMKII and NCC may weaken the interaction between NCC and NEDD4-2 effect, thereby reducing NCC degradation. The binding between filamin A and NCC may also enhance the protein’s interaction with other signaling proteins, thereby further regulating sodium and chloride transduction in DCT2.

The phosphorylation of filamin A by CaMKII results in a reorganization of the actin cytoskeleton and low basal NCC activity at the luminal membrane of a DCT2 cell. A deficiency in CaMKII or the pharmacological inhibition of CaMKII by KN93 blocks the phosphorylation of filamin A, resulting in a dense actin cytoskeleton and an accumulation of the active phosphorylated form of the NCC at the luminal membrane.

Tokonami et al.’s [32] data support the notion of a role for uromodulin in functional heterogeneity along with the DCT: Uromodulin (UMOD)-/-mice show less phosphorylation NCC (pNCC) in DCT1 and increased pNCC in DCT2 concomitantly with DCT2 extension. When exposed to chronic distal salt loads (furosemide), UMOD-/- mice displayed a severely diminished ability to increase NCC phosphorylation and lacked the ability to upregulate the pNCC in DCT1 and the structure in DCT2 regions compared to UMOD+/+ mice. Various studies indicate the substantial plasticity of the DCT. Lalioti et al. [33] used the RPCI-22 female 129S6/ SvEvTac mouse that was screened by the hybridization to radiolabeled PCR-amplified probes of the mouse Wnk4 gene to demonstrate that the DCT plasticity was essentially driven by the activity of NCC, with corresponding alterations in electrolyte and blood-pressure control. Loffing et al. [34] used adult male Wistar rats to investigate whether cell proliferation contributes to the salt-load-induced hypertrophy of distal tubules. The data demonstrated that the DNA synthesis rate markedly increased in the DCT and the following segments in vivo varied in parallel with changes in their salt-transport activity and increased DNA synthesis; thus, cellular proliferation is probably a component of the structural response of nephron segments following increased salt-transport activity. The question of whether these structural changes result from the lack of uromodulin acting as a trophic factor for the DCT1 or reflect changes in NCC activity requires further investigation. 

## 3. Ion Regulation

### 3.1. K

Patients with Gitelman syndrome (an autosomal recessive disorder associated with hypotension, hypokalemic metabolic alkalosis, and hypocalciuria) suffer from hypokalemia, while patients with PHAII (a syndrome featuring hypertension and hyperkalemia) suffer from hyperkalemia, suggesting that NCC is also essential for K homeostasis [35]. Dietary K^+^ intake is negatively correlated with BP, and several studies have shown that NCC phosphorylation is regulated by dietary potassium intake, with a high-potassium diet inhibiting and low potassium intake increasing NCC activity [35]. 

Vitzthum et al. [36] took sodium-filled mice as experimental subjects, fed different experimental groups food with different potassium contents, measured the blood pressure of mice 8 days later, and found that the low-potassium-intake mice and the high-potassium-intake mice both demonstrated increased blood pressure with sodium retention. NCC and its positive regulating enzyme, SPAK, were up-regulated under the low-potassium diet, which suggests that low K^+^ leads to increased NCC activity and increased renal-sodium-reabsorption efficiency.

Taniyama et al. [37] pointed out that the possible reason why increased renal sodium reabsorption causes hypertension is that after the increase in renal sodium reabsorption, the body inevitably encourages the increase in renal water reabsorption in order to maintain the steady state of sodium-ion solubility in plasma, while the intravascular increase in blood volume is an increase in the return of venous blood to the heart, thereby increasing cardiac output, causing an increase in blood pressure.

### 3.2. Mg

In recent years, our understanding of the molecular mechanisms of renal magnesium (Mg^2+^) processing has greatly improved. 

Ferdaus et al. [38] found that dietary Mg^2+^ restriction decreased the abundance of total and phosphorylated NCCs in mice. In mice lacking tubular NEDD4-2, Mg^2+^ restriction no longer reduced total NCC or phosphorylated NCC, suggesting that Mg^2+^ reduction increases NEDD4-2 activity and, thus, blood pressure.

Van der Wijst et al. [39] showed increased renal TRPM6 mRNA and protein expression in mice fed an Mg-deficient diet, whereas an Mg-enriched diet upregulated TRPM6 mRNA levels in the colon, and the inhibition of NCC with thiazide diuretics downregulated TRPM6 expression.

### 3.3. Ca 

Bazúa-Valenti et al. [40] proved that extracellular calcium regulates calciuria by acting on calcium-sensing receptors (CaSR) in the basolateral membrane of the thick ascending branch of the loop of Henle (TALH), thereby reducing calcium reabsorption at the expense of the apex absorption of NaCl. CaSR is also expressed in the apical membranes of distal convoluted tubules. Here, they used Xenopus laevis oocytes in vitro and C57BL/6 male-mouse in vivo models to show that the stimulation of CaSR induces the activation of NCC through a pathway involving the protein kinase C (PKC)-induced activation of the KLHL3-WNK4-SPAK pathway, ultimately phosphorylating NCC and increasing its activity, thereby increasing its activation and regulating blood pressure. 

## 4. NCC Is Affected by Drugs

Thiazides, as NCC blockers, reduce peripheral vascular resistance as the main mechanism of its antihypertensive effect. The chronic treatment of salt-sensitive animals with hydrochlorothiazide (HCTZ) also attenuates salt-induced renal injury, including podocyte injury, peritubular capillary loss, tubular atrophy, macrophage infiltration, and interstitial fibrosis [41]. 

Tutakhel et al. [42] demonstrated that calcineurin inhibitor (CaN) treatment increased total NCC (tNCC) and pNCC abundances in urinary extracellular vesicles (uEVs) isolated from kidney transplant recipients. The uEVs are nanosized membranous vesicles released from all cells lining the nephron. Alterations in the expression of different proteins present in the epithelial cells of DCT, including NCC, are reflected in the composition of uEVs. Their studies showed that Tacrolimus (Tac)-treated hypertensive kidney-transplant recipients respond to chlorthalidone. Blood-pressure responses are associated with pNCC abundance in uEVs [42].

Mu S et al. [20] used male C57BL/6j mice, the Dahl model, and deoxycorticosterone acetate-salt (DOCA-salt) rats to demonstrate that β2 adrenergic receptor (β2-AR) stimulation in salt-loaded mice leads to WNK4 downregulation, NCC activation in the DCT, salt retention, and increased blood pressure.

Poulsen et al.’s [43] studies showed that the salbutamol stimulation of β2-AR is an effective and rapid activator of NCC, and it mimics sympathetic hyperactivity. In vivo, chronic Salbuterol infusion increases blood pressure; therefore, it may be a risk factor for hypertension in humans [43]. 

Kidoguchi et al. [44] showed the natriuretic effect of Azilsartan by inhibiting the WNK4-SPAK-NCC pathway through sympathetic inhibitory activity in a model of adenine-induced chronic renal failure. 

Wang et al. [45] used low-dose L-NAME administration on mice and confirmed that L-NAME activates NCC through the oxidative-stress–pSPAK pathway to impair urinary sodium excretion and induce blood-volume expansion, ultimately leading to salt-sensitive hypertension.

## 5. Other Adjustments

### 5.1. Sympathetic Nervous System

The discovery of the antagonistic effect of Franco Puleo on the α1-adrenoceptor further supports the existence of renal sympathetic nerve-mediated (SNS) WNK4/SPAK/OxSR1 pathway that regulates NCC activation in the long term [46].

The study by Fujita et al. [47] confirmed that the mineralocorticoid receptor (MR) and the glucocorticoid receptor (GR) are stimulated by Ras-related C3 botulinum toxin substrate 1(Rac1) and renal SNS overactivity and participate in the activation of ENaC at NCC/ENaC and CNT at DCT2, and the cortical collecting duct (CCD) is downregulated by WNK4 through Sgk1 and NCC activation in DCT1, thus activating NCC.

### 5.2. Immune System

In CKD, Furusho et al. [48] confirmed that salt-sensitive hypertension is induced by the TNF-α-activated renal WNK1-SPAK-NCC phosphorylation cascade, which reflects the relationship between NCC and the immune system. 

Liu et al. [49] used CD8 ^+^ T cells to directly contact the distal convoluted tubules in the kidneys of DOCA-salt mice and mice injected with CD8 ^+^ T cells, resulting in the upregulation of NCC and p-NCC and the development of salt-sensitive hypertension. This provides an explanation for the involvement of adaptive immunity in the pathogenesis of renal defects during sodium treatment and salt-sensitive hypertension.

### 5.3. Hormones

It is well-known that NCC is regulated by gender dimorphism [50]. 

The results presented by Lorena Rojas-Veg [51] showed that ovarian hormones and prolactin (PRL) increase renal NCC phosphorylation, and estradiol (E2), progesterone (P4), and prolactin are active hormones during the participatory cycle, pregnancy, and lactation, all of which can upregulate the activity of NCC. 

Ava et al. [52] used a DCT-specific CA-SPAK mouse model to isolate the direct effects of NCC activation in DCT from downstream consequences in aldosterone-sensitive distal nephrons (ASDN). The cells in the ASDN are the primary sites of potassium secretion; sodium absorption through the ENaC creates a favorable driving force for potassium to be transported into the tubule lumen through the chief potassium excretory channels and ROMK. The data revealed that the autocrine pathway of PGE2-EP1 is activated by low salt transport in ASDN, which inversely couples potassium secretion in ASDN with NCC activation.

NCC activity is highly regulated by a complex signaling network. KS-WNK1 is thought to inhibit L-WNK1 kinase activity and its effect on NCC through a dominant-negative effect. WNK4 can form dimers with KS-WNK1, which may be unable to bind chloride due to the lack of kinase domain (KD) and chloride binding sites. The resulting KS-WNK1-WNK4 heterodimer is insensitive to chloride and, thus, phosphorylates WNK4. NCC interacts with the adaptor protein AP-3 to participate in lysosomal transport, angiotensin II converts WNK4 from an inhibitor to an activator of NCC, and WNK4 acts as a regulator to adapt to K+ changes and Na^+^ balance. Mutant WNK4 has a dominant negative effect on WNK3 inhibition by wild-type WNK4, and uninhibited WNK3 is shown to activate NCC through SPAK-dependent and SPAK-independent processes. KLHL3/ CUL3-based ubiquitin ligases are involved in NCC activation. Nedd4-2 is another participant that stimulates NCC ubiquitination and is inhibited by TNF-α. UMOD increases the phosphorylation ability of NCC, and Hsd11b2 participates in the activation of NCC through MR. CaMKII not only indirectly regulates NCC through filamin A but also directly phosphorylates and regulates NCC activity, and the binding between CaMKII and NCC may weaken the interaction between the effects of NCC and NEDD4-2, thereby reducing NCC degradation. GR is stimulated by renal SNS overactivity and participates in the activation of NCC. Thiazides are NCC blockers, while CaN increases tNCC and pNCC abundance, and Tac, ARB, and B2-AR regulate NCC through the WNK pathway. L-NAME activates NCC through the oxidative-stress-–pSPAK pathway. Potassium, calcium, and magnesium plasma regulate NCC through the WNK pathway. Ovarian hormones and prolactin PRL can increase NCC phosphorylation in the kidneys, and estradiol E2, progesterone P4, and PGE2-EP1 can upregulate NCC activity.

## 6. Conclusions

Although this review focused on the mechanism of NCC in relation to hypertension in CKD, renal sodium excretion depends significantly on other sodium transporters, including NHE3, NKCC2, and ENaC. Upon NCC activation, other transporters may also be affected, and an opposite compensatory response may even occur. NCC is regulated by a large and complex network structure, but there are still many unsolved problems. We expounded on the various regulatory mechanisms of NCC in relation to hypertension in CKD to summarize these insights. Hence, a better understanding of the molecular mechanisms that regulate NCC could lead to the development of drugs that target NCC regulation with fewer adverse effects and to a better understanding of and treatment for hypertension and disorders of extracellular fluid volume. As the WNK-SPAK-OSR1-NCC pathway is the main regulator of NCC, many therapeutic approaches targeting this pathway have been developed, including clinically used drugs, such as Tac, ARB, and β2-AR. So far, as we know, specific modulators for WNK or SPAK are not yet in a clinical trial. Salt intake and circulating volume vary widely, hour-to-hour, day-to-day, or seasonally, and the activation of WNK, SPAK, or NCC should be finely regulated. The development of a method to modify those signals within physiological feedback and stop their inappropriate activation is required (Figure 1).

## Figures and Tables

**Figure 1 ijms-24-00286-f001:**
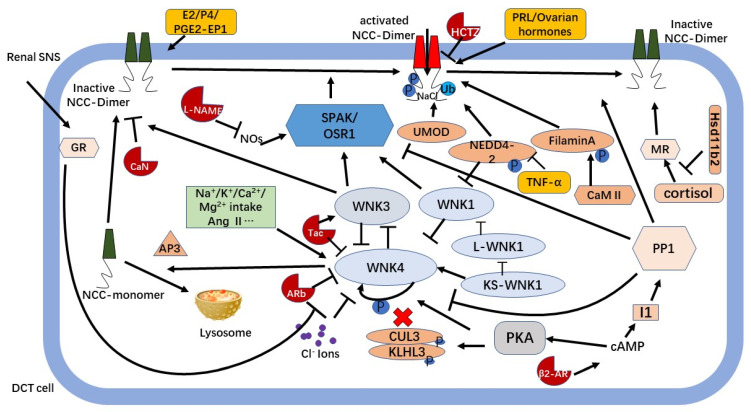
Regulation of NCC by a complex signaling network. Arrowhead lines and T-shaped lines indicate activation and inhibition, respectively.

## Data Availability

Not applicable.

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
