# Peer review of "Molecular Mechanisms of Na-Cl Cotransporter in Relation to Hypertension in Chronic Kidney Disease"

_ijms, 2022, doi:10.3390/ijms24010286_

Round 1
Reviewer 1 Report
Comments to the authors:
1. Authors should give the full name for the abbreviation of DCT, NCC, WNK, SNS, RAAS, SGK1, CNT/CD, ROMK, NKCC2, etc. when they first mention them. Also, give a full list of abbreviations.
2. Line 26: In keywords add “kinases” after “WNK”, also add “chronic kidney disease”
3. Line 288-289: give the reference from which the figure is taken.
4. Line 24-25: What do you mean by “inappropriate activation of NCC in CKD and hypertension”? What is inappropriate activation of NCC?
5. Line 62-63: rewrite and define better the aim of the study; the phrase “in CKD hypertension” is not correct to use., what do you mean by that phrase? Also, correct this throughout the manuscript.
6. Line 124: “Evidence for the effect of WNK3 on these transporters and their co-expression in renal epithelial cells supports that WNK3 is associated with NaCl, water, and blood pressure stabilization.” Which transporters do you mean, name ones. Also, give the reference for this statement.
7. Line 135: delete “As” at the beginning of the sentence.
8. Regarding sections 2., 3., and 4. of the manuscript, authors have not presented the results from cited papers in relation to the role of NCC in hypertension as well as in chronic kidney disease, which they define as the aim of their study; they should highlight every experimental model used in those studies and main results.
9. Reference Ishizawa et al. 2016 (line140), and Vitzthum et al. 2014 (line 197) are missing from the references’ list.
10. Reference “Subramanya, A.R.; Liu, J.; Ellison, D.H.; Wade, J.B.; Welling, P.A. WNK4 Diverts the Thiazide-Sensitive NaCl Cotransporter 360 to the Lysosome and Stimulates AP-3 Interaction. Journal of Biological Chemistry 2009, 284, 18471–18480, 361 doi:10.1074/jbc.m109.008185” from the reference list is not cited in the manuscript.
11. Line 77: authors have pointed out the action of NCC in hypertension, without any function of NCC in pathophysiology of CKD which is not in accordance with the title and aim of the manuscript.
12. Statements in the paper are not supported by the references. Many references are missing, provide ones at the end of the sentences in the next lines: 80, 82, 84, 87, 102, 120, 123, 128, 133, 170, 187, 192-193, 202, 240, 241, 243, 249, 278.
13. What do you mean by this text “3. NCC is affected by drugs” in line 230? Is it a new title section?
14. Sentence in lines 246-248 “Poulsen et al. studies have shown…, and it mimics sympathetic hyperactivity” is duplicate from previous one, delete it.
15. In sections 2., 3., and 4. only one reference is related to chronic renal failure (line 250-252) which is far away from sufficient. Provide more references regarding NCC function in CKD, since the aim of the manuscript is to include hypertension and chronic kidney disease.
16. Which experimental models are used in these studies in lines 108, 114, 185, 227, 243?
17. Line 284: what is “ASDN”?
18. What do you mean by this sentence “This leads to sodium retention and salt-induced development of hypertension.” in line 106-107? Connect it adequately with the text in that paragraph.
19. In lines 262-263: full name of the receptors should be out of the brackets, only abbreviations should be in the brackets.
20. Regarding section 4. Other adjustments, authors presented 4.1., 4.2., and 4.3. subsections only by two references which is a very small number, please provide more studies that are related to NCC action in pathophysiology of the CKD and hypertension.
21. In abstract authors mentioned that “in their previous studies they had investigated the mechanisms of how NCC is activated in several different models” (line 23-24). Where are the references of these studies, what experimental models have they used, and what are the results?
22. In Conclusion delete references and rewrite it in relation to the title of the manuscript and all the results that you have presented in the manuscript related to NCC action in pathophysiology of the CKD and hypertension. What is the main conclusion of this study?
Reviewer 2 Report
Authors have attempted to write a review on the regulation of Na Cl Cotransporter (NCC) in hypertension. The subject matter is of great interest to physicians and investigators working in the areas of kidney disease, hypertension, and tubular function. The following concerns should be addressed.
Major concerns
Authors are encouraged to seek the assistance of a person familiar with English grammar and composition. They are encouraged to avoid long and unwieldy sentences that are hard to understand.
Authors should give the full name of molecules along with abbreviations when they first appear in the article. A separate list of abbreviations and corresponding names would also help.
Line 12-14. Reframe sentence with regards to CKD appearing twice.
Line 86: Reference for KS-WNK1 inhibiting L-WNK1 should be provided. Authors should define the differences between KS-WNK1 and L-WNK1.
Line 93. What determines whether WNK4 acts as a positive or negative regulator of NCC? Chloride has emerged as an important modulator of WNK4 activity. This must be discussed.
Authors have missed discussing WNK bodies.
Lines 103-107. Authors should employ simple sentences. This comment applies to many other passages in the manuscript.
Line 108: the counter-intuitive action of angiotensin II on WNK4 is of interest. Authors should comment on the molecular mechanism by which Ang II inhibits the negative regulation of NCC by WNK4.
Line 163: Change to Regulation by…..
Line 166: Define Hsd11b2.
Line 173: Readers should be told how filamin A interaction with NCC regulates the activity of the latter. In the setting of NCC regulation what activates CamKII? This paragraph sheds little light on these mechanisms.
Line 185. Cellular hypertrophy involves increased protein synthesis and cell size with minimal changes in DNA synthesis and cell division.
Line 192: Define the characteristics of Gittleman syndrome. Also, define PHAII and its features in addition to hypekalemia?
Line 194. Vizthum reference is missing in the References Section.
Line 224. It is loop of Henle, and not Henry.
Line 237. Define uEVs.
Line 284. Define ASDN.
Figure 1: WNK1 is said to exist as L-WNK1 or KS-WNK1 in the kidney. Why is WNK1 shown as a separate entity from L-WNK1 and KS-WNK1? Figure is too busy. It should be simplified and divided into panels highlighting individual regulatory pathways.
Conclusion: Authors should discuss potential sites for therapeutic intervention in the WNK-SPAK-OSR1-NCC pathway to mitigate hypertension and CKD. Are there such drugs in development?
Round 2
Reviewer 1 Report
Comments to the authors:
1. The authors omit to change the title of the manuscript version 2, as they suggested, “The mechanisms of Na-Cl cotransporter in relation to hypertension in chronic kidney disease”; Suggestion: instead of “the” at the beginning of the title, here you can put “Molecular mechanisms of …”.
2. Line 15, typo error: add “disease” after “cardiovascular (CV)”
3. Line 399, grammatical error: delete “in”
4. Line 400, grammatical error: instead of “the modulating” change to “the activation (or action) of”
